# Integrating Human Barriers in Human Reliability Analysis: A New Model for the Energy Sector

**DOI:** 10.3390/ijerph19052797

**Published:** 2022-02-27

**Authors:** Dina Guglielmi, Alessio Paolucci, Valerio Cozzani, Marco Giovanni Mariani, Luca Pietrantoni, Federico Fraboni

**Affiliations:** 1Department of Educational Science, University of Bologna, Viale Filippo Re 6, 40126 Bologna, Italy; alessio.paolucci2@unibo.it; 2LISES—Department of Civil, Chemical, Environmental and Materials Engineering, University of Bologna, Via Terracini 28, 40131 Bologna, Italy; valerio.cozzani@unibo.it; 3Department of Psychology, University of Bologna, Via Berti Pichat 5, 40126 Bologna, Italy; marcogiovanni.mariani@unibo.it (M.G.M.); luca.pietrantoni@unibo.it (L.P.); federico.fraboni3@unibo.it (F.F.)

**Keywords:** human reliability, HRA, human error probability, human factors, safety barriers

## Abstract

Human reliability analysis (HRA) is a major concern for organizations. While various tools, methods, and instruments have been developed by the scientific community to assess human error probability, few of them actually consider human factors impact in their analysis. The active role that workers have in shaping their own performance should be taken into account in order to understand the causal factors that may lead to errors while performing a task and identifying which human factors may prevent errors from occurring. In line with this purpose, the aim of this study is to present a new methodology for the assessment of human reliability. The proposed model relies on well-known HRA methodologies (such as SPAR-H and HEART) and integrates them in a unified framework in which human factors assume the role of safety barriers against human error. A test case of the new method was carried out in a logistics hub of an energy company. Our results indicate that human factors play a significant role in preventing workers from making errors while performing tasks by reducing human error probability. The limits and implications of the study are discussed.

## 1. Introduction

Statistics show that human error is one of the primary causes of failure in a variety of working contexts, such as in the nuclear [1], chemical and petrochemical [2], maritime [3], and aviation sectors [4]. Human reliability analysis (HRA) is a systematic technique to assess human error probability and has been widely used in various industries for enhancing the safety and reliability of complex socio-technical systems. Several human reliability analysis (HRA) methods have been developed to predict human error associated with working performance. They estimate the probability of human error based on workers’ actions and decisions and use system engineering and cognitive and behavioral sciences to assess and evaluate human activities’ impact on safety and reliability systems [5].

Recently, Hou et al. [6] conducted a comprehensive bibliometric analysis of the HRA field, identifying blind spots in the literature and suggesting potential future research directions. According to the study results, traditional HRA methods are not suitable for today’s working environments due to the transformation of tasks and systems. For this reason, the authors suggest focusing on the development of advanced HRA models that could overcome such limitations.

Despite the success of existing HRA models in increasing safety in many organizations, they still do not adequately consider psychosocial barriers. For instance, among currently available HRA methods, the Standardized Plant Analysis Risk Human Reliability Analysis (SPAR-H) and the Human Error Assessment and Reduction Technique (HEART) are two of the most widely used and acknowledged [7,8]. Both techniques share the assumption that human errors can be predicted by predetermined factors, such as task typology, context, and working conditions, shaping human performance and determining the probability of an error occurring. While these methods proved their value in predicting human error, they usually fail to consider the active role of workers (e.g., their behaviors and attitudes) and the psychosocial dynamics involved in working performance. In this regard, the crucial role played by human factors in safety was proven by several studies and across different working sectors (for a review, see [9]). In line with this assumption, French et al. [10] pointed out that to further extend and improve HRA methods, workers’ cognitive and affective processes and organizational factors influencing performance (e.g., safety culture, safety climate) should be included. Moreover, the authors proposed to consider psychosocial “barriers” that play a fundamental role in preventing and protecting workers from adverse events at work.

Therefore, the present paper aims to extend current HRA methods by including psychosocial factors and barriers in a revised framework to analyze human errors in energy companies. The proposed approach is in line with a new scientific trend focused on the inclusion of human factors in quantitative risk analysis [9]. However, the developed model distinguishes itself from previous ones in its primary focus on human factors and its applicability to various tasks carried out in the energy sector. The present paper relies on a novel approach that assesses how organizational and human factors can affect workers’ safety performance and the probability of human error.

## 2. Literature Review

### 2.1. Traditional HRA: SPAR-H and HEART Methodologies

The overall assumption of SPAR-H, shared with many other HRA methods, is that the occurrence probability of a human error is strictly associated with the typology of the task to be performed and the presence of factors that can positively or negatively influence human performance. Consequently, the method consists of two main components through which it is possible to quantify the human error probability (*HEP*): task typology (diagnosis or action tasks) and performance shaping factors (PSF), which can be described as the sum of variables belonging to the individual, the environment, and the organization or the task (all of which can influence human performance). Although often with some adaptations, the SPAR-H methodology has been used successfully in various energy companies to assess *HEP*. Mirzaei Aliabadi et al. [11], for instance, integrated Bayesian networks to SPAR-H for assessing *HEP* of pipeline inspection gauges operations in a gas transmission plant. Abreu et al. [12] successfully used the SPAR-H to assess *HEP* in relation to cascade events in power grid management. However, they integrated the methodology with stochastic simulations using the Monte Carlo methods to increase its effectiveness. Van de Merwe et al. [13] applied the SPAR-H to a normal-operations scenario for a managed pressure drilling concept, integrating the systematic human error and prediction approach (SHERPA) to gain more insight into the qualitative aspects of the operations.

Although SPAR-H has been validated across different industrial fields, scientists and practitioners have highlighted various methodological issues. For example, Liu et al. [14] recently showed that PSFs are unclearly defined, resulting in the uncertainty of human reliability analysis. They thus proposed an expert-based modification approach for redefining the PSFs based on four criteria to reduce the overestimation of human error probabilities: less overlap, hierarchy, flexibility, and digitalization. Another issue was highlighted by Wang et al. [15], according to which dependencies between PSFs have not been analyzed and need to be considered to improve SPAR-H methodology further. More specifically, the authors found the existence of mediating and moderating effects between PSFs [15]. In line with this limitation, new HRA methods, such as the Systematic Human Reliability Analysis (SHRA), were developed to integrate dependencies between PSFs in the *HEP* estimation process [16].

The HEART methodology [8] is based on the principle that human reliability depends on the typology of the task to be performed and on the presence of external conditions that affect human performance. HEART methodology is thus based on two major components: generic task types (GTT) and error producing conditions (EPCs). The nine GTT identified in the model can be described as categories that allow for identifying of the peculiarities of a given task, and they are used to determine the predetermined probability of human failure. The 38 EPCs included in the HEART method represent the sum of factors hindering human performance, and each is associated with a multiplier that determines the impact of the variable on human reliability. Over the past 30 years, HEART methodology has been implemented and validated across many industrial fields and sectors, such as nuclear [17], aviation [18], railways [19], maritime [20], and energy industries [21].

### 2.2. Human Factors as Safety Barriers

Although traditional HRA methods such as SPAR-H and HEART proved their value in predicting *HEP*, they usually neglect the complexity of human and organizational factors shaping the overall safety performance and the active role of operators and teams in preventing human error and maintaining safety. For instance, while non-technical skills proved to be relevant for safety performance [22], HRA methods usually do not consider non-technical skills such as situation awareness or safety communication shaping operators’ actions.

Human and organizational factors in safety are not always a top priority for employees. In some sectors, workers and managers are scarcely aware of the role of human factors in their job performance, and there is not adequate communication between management and workers around those topics. Karanikas et al. [23] found that the longer employees worked for the same company, the more they realized that errors were caused by people interacting with their local working environment (e.g., equipment, tools, coworkers) and unsafe actions were not merely a result of their performance.

On the contrary, in some sectors, the awareness of how local factors affect human performance becomes more profound, and the critical role of understanding human factors and the importance of safety culture is becoming much more topical. Corrigan et al. [24], for example, showed an increasing awareness of human factors and a move towards a positive safety culture in the maritime sector.

Although a clear and unanimous definition of safety barriers is still missing, these are usually intended as the sum of technical and non-technical means planned to prevent, control or mitigate undesired events or accidents [25]. Safety barriers can be classified on the basis of their characteristics and functions. For example, the well-known classification proposed by Hollnagel [26] distinguishes between physical, functional, symbolic and incorporeal barriers, while [27] categorizes the barriers as engineered, organizational, and human. In our model, we propose to gauge human factors through the concept of a “barrier”.

The present study draws from two of the most widespread barrier models, the Swiss cheese model [28] and the bow-tie model [27], to understand how barriers carry out their protective role and their causal dynamics with human errors.

The pioneering Swiss cheese model was initially developed by Reason in 1990 to explain the causes of workplace accidents. According to the model, the function of a barrier is to protect workers from potential threats (i.e., hazards), which, if not controlled, can result in accidents. The model posits that accidents happen when an alignment in the holes of the layers of defense (the barriers) causes the hazard to pass through the defensive lines [28]. Despite its popularity, the Swiss cheese model presents a few limitations. First, while the model proved its value and utility from a scientific viewpoint, it does not provide practical means to effectively analyze accident dynamics at work [29]. Furthermore, the Swiss cheese model does not clearly define barriers [30], thus hindering the possibility of being able to accurately identify those factors that may prevent or protect from hazards.

In line with these developments, Hu [31] identified three levels of barriers, respectively: hardware and human, organizational management and supervision, and safety culture. It is possible to distinguishes them as direct and indirect barriers according to the modality through which they play their protective roles. Only first level barriers are considered as direct barriers since they hinder the spread of hazards. In contrast, second and third level barriers indirectly play their protective roles by influencing the defensive quality of direct barriers. For example, safety compliance needs to be considered a direct barrier since it directly affects operators’ behavior. Safety culture, or safety climate, is not directly connected with workers’ behaviors, but is indirectly associated with risk exposure since it is related to the workers’ attitudes towards safety [31].

Traditionally, the Bow-tie Model has been used to explain and visualize accidental dynamics in sectors such as aviation and the energy industry [32]. It describes a linear process, according to which potential causes of accidents may become the precursors of a critical event that in turn may evolve, generating several alternative outcomes (e.g., damage). In this model, barriers represent the sum of factors that prevent the potential causes from triggering the critical event and protect workers from the more severe consequences of the latter [33,34]. In other words, barriers simultaneously play a preventive role—since they hinder the manifestation of critical events—and a protective role, as much as they protect from the consequences of critical events. Accordingly, the proposed barriers are considered relevant factors in estimating the probability of human errors.

## 3. Development of the Methodology

In the following sections, we will describe how the new methodology has been developed, adapting and integrating previous HRA models.

The present model was developed to assess *HEP* in a multinational energy company operating in 68 different countries (26 countries in Europe, 6 in the Americas, 14 in Africa, and 22 in Asia and Oceania) with more than 31,000 employees. The HRA method was developed conforming to the theoretical framework discussed above, considering the organization’s characteristics and with the participation of the company’s health, safety and environmental (HSE) managers, researchers, and field experts.

The complexity and specificity of the methodology development required us to choose one organization that was sufficiently representative of the entire energy sector. The model includes indicators of safety climate and culture that can influence multiple other factors, highly increasing the complexity of the assessment. Thus, we decided to choose a single organization that would allow us to carry out all of the activities. The developed model can be considered a prototype to be further tested and extended into other organizations. Therefore, the selected organization satisfies a series of requirements that can increase the generalizability of the model to other contexts, specifically: the large number of countries in which it operates; the high number of offices and operational sites; the size of operations; and the large workforce and the wide variety of tasks.

To overcome SPAR-H and HEART limitations and to meet organizational contextual needs, an original approach based on the combination of these methodologies was developed. A new assessment methodology including the strong points of both methods was thus obtained. Both methods estimate human reliability according to a given task to be performed. However, while SPAR-H distinguishes between action and diagnosis tasks, HEART provides the possibility to choose between nine GTT. Working tasks in the energy company context are characterized by a high level of complexity and cognitive load. Consequently, determining whether a task is diagnostic or action could be complex and would not allow a specific distinction among the tasks under analysis. Thus, the proposed method introduced a revision of the GTTs of the HEART methodology. The need for a redefinition of GTT derives from Williams and Bell’s study [35], which revised the scientific evidence on the implementation of HEART and found empirical support for the construct validity of seven out of nine of the original GTT. In line with the study, the model was based on the seven GTT [35], which were revised and modified according to the organizational context (see Table 1). In line with the HEART methodology, each GTT was then associated with a nominal error probability (NEP).

The second step required to assess *HEP* involves the identification of factors affecting performance. While SPAR-H recognizes the role of 10 main PSF, HEART methodology identifies 38 EPC. Due to the necessity of contextualizing the method in the energy company, a revised version of SPAR-H’s PSFs was used in this study since they represent the most efficient categorization of factors affecting performance [36]. In line with [36], only the PSFs most impactful on performance were included. Moreover, “teamwork” and “adequacy of the organization” dimensions were not included to reduce the conceptual overlap with human barriers. More specifically, these PSF were excluded since their goal is to indirectly measure non-technical aspects of safety, such as the workers’ ability to work together or the quality of organizational measures in terms of safety. While these PSFs are indeed relevant for assessing human error, in the proposed method they were replaced with specific barrier dimensions, which allow a more precise and detailed evaluation of both teams and organizations. The PSFs included in the model are presented in Table 2.

To evaluate the impact of PSFs on human reliability, multipliers of the error probability were associated with each of them, based on evaluation levels. These have been identified through the review of the SPAR-H methodology carried out by Laumann et al. [36], and they have been modified according to the contextual needs of the energy industry [37]. Table 3 describes the multipliers proposed for each PSF.

### 3.1. Human Barriers Dimensions

In the proposed model, human error probability is the product of the interaction between three main elements: the typology of the task to be performed (GTT), the conditional factors affecting performance (PSF), and human factors (human barriers), which are the sum of human and organizational assets of safety preventing workers from performing errors. Figure 1 graphically shows the interaction between the elements of the model.

To identify the barriers to be included in the model, an initial pool of 20 psychosocial dimensions relating to both safeguard and direct barriers were identified through a literature review focused on human factors related to safety in organizations [38,39]. The distinction between direct and safeguard barriers was made according to the specific role they play in preventing human error. Although a clear distinction between human barrier categories could not be retrieved from scientific literature, the proposed classification is based on a shared assumption among scientists, according to which the direct involvement of barriers in error prevention varies on the basis of their role and characteristics [31]. While direct barriers are supposed to influence human behavior directly, safeguard barriers represent those layers of defense that determine the quality and availability of direct ones.

The content validity of barriers was established by applying Lynn’s method [40]. A total of five judges were provided with an evaluation sheet through which they were instructed to assess whether the identified dimensions could represent safety human barriers and whether they belonged to a direct or safeguard category. The judges were three full professors with an average of 14 years of experience in the industrial-organizational-psychological field and included two PhD students.

The inclusion criteria were the following:(1)Dimension’s relevance as a barrier: To assess whether the dimension could be considered as a human safety barrier, the judges were asked the following question: “To what extent do you believe that the psychosocial dimension can positively influence workers’ safety behavior?”(2)Barrier category: To assess whether the dimension could be considered as a direct or safeguard barrier, the judges were asked the following question: “To what extent do you believe that the psychosocial dimension directly prevents workers from making errors during task execution?”

A 4-point Likert scale was provided to rate both questions ranging from: 1 = irrelevant; 2 = somewhat relevant; 3 = quite relevant; 4 = extremely relevant. The content validity index for the first criterion was then computed as the number of scores equal to or greater than three for each question (divided for the number of judges).

In line with Lynn’s suggestions [40], only those dimensions with indexes equal to 1.0 were retrieved in the model. In order to place the identified dimensions in the respective categories of barriers (direct or safeguard), the average values of the scores of the second question were calculated. Dimensions with scores equal to or higher than 2.0 were assigned to the direct barrier category, while those with rates lower than 2.0 were considered safeguard barriers. Table 4 provides a list of the dimensions identified for each barrier typology.

To assess the quality of direct and safeguard barriers, an evaluation checklist was prepared. Multiple key indicators representing core aspects and peculiarities of the barriers were identified for each barrier dimension to quantify their quality, concerning their ability to protect operators at work and to prevent errors. For example, the following indicators were defined for safety task performance: “Does the operator work safely while performing the task?”; “Does the operator follow safety norms and procedures related to the task?”.

To assess the quality of barriers, indicators in the checklist were associated with different evaluation levels, to which scores ranging from 0 to 1 were assigned. Table 5 shows levels and scores for the assessment of human barriers.

A barrier score, *BS**_d_*, is calculated for each relevant dimension, as the mean of the scores of the indicators (*I*) belonging to the specific dimension:(1)$$BS_{d}=\frac{\sum_{j=1}^{n_{d}}I_{j}}{n_{d}}$$
where *n_d_* is the total number of indicators considered for dimension *d*. A global barrier score for both direct (*DBS*) and safeguard barriers (*SBS*) is then calculated as the mean of the barriers scores belonging to each of the two categories:(2)$$DBS=\frac{\sum_{d=1}^{m_{D}}BS_{d}}{m_{D}}$$
(3)$$SBS=\frac{\sum_{d=1}^{m_{S}}BS_{d}}{m_{S}}$$
where *m_D_* is the total number of dimensions considered for direct barriers and *m_S_* is the total number of dimensions considered for safeguard barriers. Global scores represent the overall quality of both direct and safeguard barriers, and they are used in the *HEP* estimation process, which is described in the following.

### 3.2. Estimating HEP including Barriers in the Analysis

The assessment process of *HEP* is thus carried out in six main steps: (1) determination of the NEP value of the task under analysis; (2) calculation of PSFs multipliers; (3) calculation of direct barriers score; (4) calculation of safeguard barriers score; (5) calculation of conversion coefficient of direct and safeguard barriers; (6) calculation of integrated *HEP*. The detailed assessment process, including the specific sub-steps of the procedure, is illustrated in Figure 2.

The first step of the *HEP* assessment requires analyzing the task to be performed and selecting the GTT. Once the GTT is chosen, the assessor identifies the NEP value associated with the specific category of GTT.

The second step involves the evaluation of the PSF associated with the task. To achieve this goal, the assessor may use task observation or interviews to attain all of the information required to estimate the impact of performance shaping factors on human reliability. Once the assessment is carried out, the method requires the identification of the multipliers associated with each PSF and the estimation of the overall impact of PSFs on the probability of human failure through the calculation of the value of the PSF composite (*PSFC*) as the product of the multipliers of the PSFs:(4)$$PSFC=\prod_{j=1}^{n}M_{j}$$
where *M_j_* is the multiplier selected for the *j*-th PSF and *n* is the total number of PSFs considered.

The third and the fourth steps of assessment involve the evaluation of the quality of direct and safeguard human barriers, respectively. The assessor can use multiple sources of information, such as observation and interviews, or/and surveys, to assess the values of the indicators (I) defined for each barrier dimension (see Table 4 and Table 5). Once the indicators are evaluated, the method requires estimating the mean scores of dimensions and global scores of both direct and safeguard barriers by Equations (1)–(3).

In the fifth step, the impact of human barriers on *HEP* is determined. Global scores of both direct and safeguard barriers need to be converted into new measures representing their impact on human error probability, namely the conversion coefficients. The latter are conceived as multipliers of the *HEP* (on par with PSF), positively or negatively influencing the occurrence probability of an error. Table 6 and Table 7 report the conversion coefficients obtained for both direct (*CCDB*) and safeguard barriers (*CCSB*).

As shown in the tables, in line with the assumptions of barriers models [25,27] the final *HEP* is supposed to vary according to the quality of barriers, which can play a preventive role against safety threats, thus decreasing the occurrence probability of errors. In other words, the better the quality of the barriers, the lesser the probability of human error. On the contrary, the poor quality of the barriers corresponds to an increased *HEP*.

The conversion coefficients for direct and indirect barrier scores reported respectively in Table 6 and Table 7 were estimated through expert judgment. Five experts with an engineering background with more than ten years of experience in organizational safety were asked to express their opinion about the impact of barriers on *HEP* based on the global scores of direct and safeguard barriers. Consulted experts were provided with evaluation sheets through which they were instructed to assign a score representing the expected impact of both typologies of barriers on *HEP*. The average values of the scores were calculated to determine the conversion coefficients.

The final step of the procedure (Step 6) involves the assessment of *HEP*:(5)$$HEP=\frac{\left(NEP\cdot PSFC\cdot CCDB\cdot CCDS\right)}{NEP\left(PSFC\cdot CCDB\cdot CCBS-1\right)+1}$$

Following the SPAR-H methodology, the probability of human error is the product of the nominal task probability and PSF multipliers. In line with this principle, barriers scores play the same role as PSF multipliers since they impact the probability of human failure.

## 4. Test Application of the Model

In the following section, a test application of the model is summarized. The test case was carried out in a logistics hub of an energy company. The task selected for the test is lifting operations at a dock. The test was performed by observing a full-scale routine lifting operation during normal functions of the hub.

### 4.1. Method

Two assessors were in charge of carrying out the test. The information needed to fill in all of the checklists were gathered preliminarily through three different methods:

Documental analysis of organization chart, work permit documents, area risk information, improvement plan, coordination sheets, and schedules. Assessors also had the opportunity to inspect documents regarding policies, procedures, and occupational safety internal reports;

Semi-structured interviews with supervisors, managers responsible for operations, site managers, and workers directly carrying out the selected task. A total of 11 online and 10 in-person interviews, lasting around one hour each, have been conducted;

One full day on-site direct observations of workers carrying out the task.

### 4.2. Context and Task Description

The logistics hub premises, in which the test took place, served as a hotspot for incoming and outgoing supply vessels from/to offshore gas rigs. Daily, two to three supply vessels load or unload materials and supplies at the dock, using onshore cranes. On the dock, routine operations are governed by established procedures with a one-off generic work permit. No toolbox talks or special (safety-related) briefings are given for this type of activity. Each evening, the contractor foreman receives the vessel arrival schedule for the next day, and the morning after coordinates with the dock supervisor to schedule the daily activities. Special inspections and safety briefings take place in case containers with hazardous substances are loaded or unloaded (e.g., supply of diesel fuel for emergency generators, solvents, inhibitors).

It may be concluded that, unless hazardous substances are loaded or unloaded, the task to be performed can be classified as a routinary task, requiring a low level of experience and training. Accordingly, the GTT category chosen for this task was “D”.

A contractor team was responsible for carrying out the operations. The group consisted of a foreman (who supervised and actively participated in the slinging and unhooking of loads), a crane operator, and two/three other slingers (who slung, unhooked, and moved loads using push-poles). The workers’ age range was between 30 and 50 years old, and they all had several years of experience in loading/unloading operations. From the interviews and observations, it was possible to notice that the team was well established and demonstrated particular fellowship and teamwork capabilities.

During the assessment, it was possible to observe the loading on a vessel of four medium/small foodstuffs containers, a cage of metallic materials, and a liquid nitrogen tank. The goods arrived at the dock on a semi-tractor-trailer truck. The material was unloaded from the truck by a forklift (the forklift driver and equipment were part of a different contracting company) and placed in a dedicated area near the crane, which was previously closed with barriers and jerseys by the lifting team. The assessors had the opportunity to interview the foreman of the contractors and a slinger before observing the operations. The operations took place on a sunny hot day in July, with an outside temperature of 37 °C.

During the observations, it was possible to notice outstanding professionalism and attention to the operators’ safety. However, the following unsafe condition occurred: a container arrived with an incorrectly mounted sling which caused the container to oscillate dangerously when lifted. The operators immediately noticed the problem and tried to disassemble the sling and then reassemble it correctly. After several attempts and a few tens of minutes, however, the operators realized that it was impossible to fix the sling in a reasonable amount of time. According to procedures, the load was not to be lifted using the crane. As stated by organizational procedures, the safest course of action would have been to stop work, request the container to be loaded back onto the truck, and returned to the company to have the sling fixed. Since it was a refrigerated container of foodstuffs, the foreman decided on a different procedure. The load was lifted and loaded on the vessel to connect to the vessel’s power supply, avoiding the deterioration of the contents. The responsible company was then called to fix the problem at a later stage (but before shipping the cargo offshore). An observer asked the safety supervisor, who was present during operations, to stop the lifting given the presence of an unsafe condition. However, the supervisor allowed the lifting operation to be completed before taking action.

### 4.3. Results of the Test

Figure 3, Figure 4 and Figure 5 report the evaluation of performance shaping factors, direct barriers, and safeguard barriers, respectively, obtained using the developed checklists.

Two PSFs resulted critical in the assessment: working context and procedure. The working context was evaluated as an “adverse condition” as it was an extremely hot day, and evaluators were firmly convinced that this condition had a significant influence on the decision of the contractor team to act differently from the organizational procedure. The team had to wear protective suits and equipment, highly aggravating the perceived temperature. Furthermore, the hot weather could have influenced the decision since the team was worried about the degradation of the perishable goods in the container that should be refrigerated. Thus, the PSFC was calculated as shown in Table 8.
PSFC=0.1×1×1×0.1×5.0×0.5×10.0×1=0.25

Regarding safety barriers, the scores obtained (see Figure 4 and Figure 5) indicate an overall positive performance of both direct (DBS = 0.77; CCDB = 0.2) and safeguard barriers (SBS = 0.76; CCSB = 0.2). However, two key dimensions showed some issues: *compliance with safety norms* and *procedures and safety leadership*. Regarding workers’ compliance towards safety, during the observation period, several unsafe behaviors were noticed. According to safety procedures, workers should have stopped the activity when they realized that the container could not be safely lifted. Nevertheless, workers did not stop the activity (first unsafe behavior), then decided to lift the container anyhow (second unsafe behavior). Coherently with this evidence, the key indicators of the barrier were scored as shown in Table 9.

Finally, the *HEP* was calculated as showed in Table 10.
(6)$$HEP=\frac{\left(0.02\times 0.05\times 0.2\times 0.2\right)\;}{0.02\times \left(0.05\times 0.2÷0.2-1\right)+1}=0.004\%\;$$

## 5. Discussion

Historically, HRA methods have focused their attention on the set of factors that can contribute to increasing the probability of human failure. This approach has been tested and validated in many industrial fields and across different organizational contexts. However, current HRA methods fail to consider the role of psychosocial factors and how they may shape their own safety performance [41]. Furthermore, these methods are not suitable to analyze the probability of human error in current working contexts due to the digitalization of tasks and systems [6]. Therefore, the new HRA method presented in this paper aims at including the psychosocial dynamics involved in the safety performances of workers [42,43]. To achieve this goal, psychosocial factors are included in the model as safety barriers, which are supposed to play a preventive role against the sum of external factors, leading to errors and mistakes.

The proposed model addresses the recent challenges of HRA methodologies and quantitative risk analysis (QRA) to quantify the impact of the human factor on safety at work [9]. Most of these models share the assumption that human factors play a fundamental role in terms of safety, and they are developed to analyze their impact in terms of human reliability [44,45,46].

Current models mainly address the analysis of the impact of human factors on specific tasks or events. Zhen et al. [9] pointed out that most of these models were developed to estimate the probability of loss of containment from process equipment in offshore installations. The model developed is less focused on specific tasks and can be applied to most of the tasks carried out in the energy and the process industry, offshore and onshore. In addition, it attempts to overcome the main limitations of traditional methods. The GTTs classification was used to improve the contextualization of the tasks, providing a general taxonomy of task types and their related nominal error probability. PSFs were applied to analyze the set of factors which are considered to have a major influence on the overall safety performance.

Human reliability is a significant concern in most organizational contexts. Therefore, assessing which factors may impact human performance has become a priority to identify critical areas of intervention and improve workplace safety. The developed approach provides new insights into the role of human factors as safety barriers. Therefore, the results obtained should be thus considered an evolution of our understanding of how safety performance is shaped by contextual and psychosocial factors. Moreover, it represents a useful analytic tool to assess human reliability, analyzing well-acknowledged factors that can affect human reliability and psychosocial dynamics. This allows us to include the positive role that human factors play regarding safety performance.

### Limitations and Further Research

The new HRA method developed still has some limitations. Firstly, to the best of the authors’ knowledge, safety barriers were seldom included in the analysis of human error in previous studies. Thus, this innovative feature of the methodological approach proposed needs to be further tested to establish its general value in predicting *HEP*. Future work should consider more extended testing of the model, also considering other working environments.

Secondly, the safety barriers considered were chosen according to the opinion of a limited group of experts. However, other human factors may affect human performance. Future work should aim at further expanding the model, including different psychosocial factors as safety barriers.

Thirdly, the estimation of *HEP* needs to be grounded in validated mathematical approaches. Implementing the methodology on larger samples will allow evaluating the impact of each barrier on *HEP* more.

## Figures and Tables

**Figure 1 ijerph-19-02797-f001:**
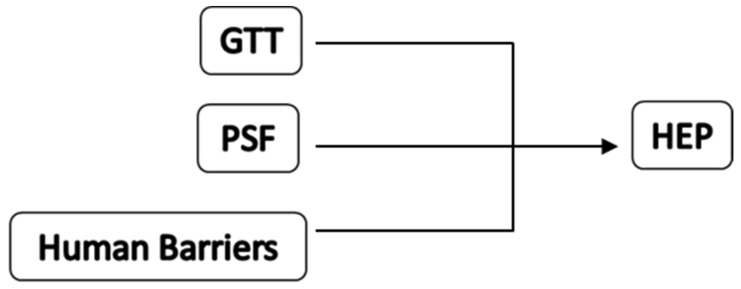
The revised human reliability analysis model.

**Figure 2 ijerph-19-02797-f002:**
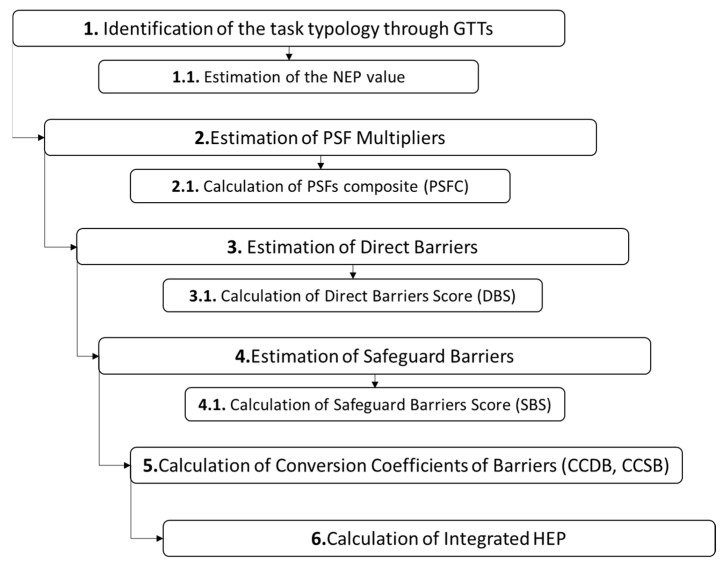
Summary of the assessment process.

**Figure 3 ijerph-19-02797-f003:**
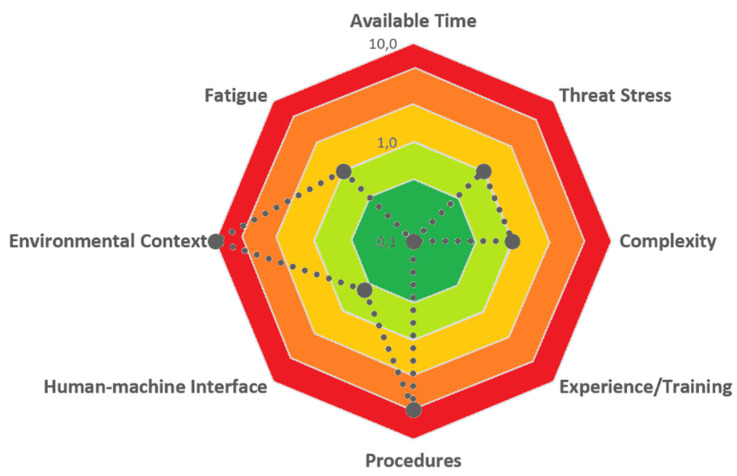
PSF Scores obtained in the test case.

**Figure 4 ijerph-19-02797-f004:**
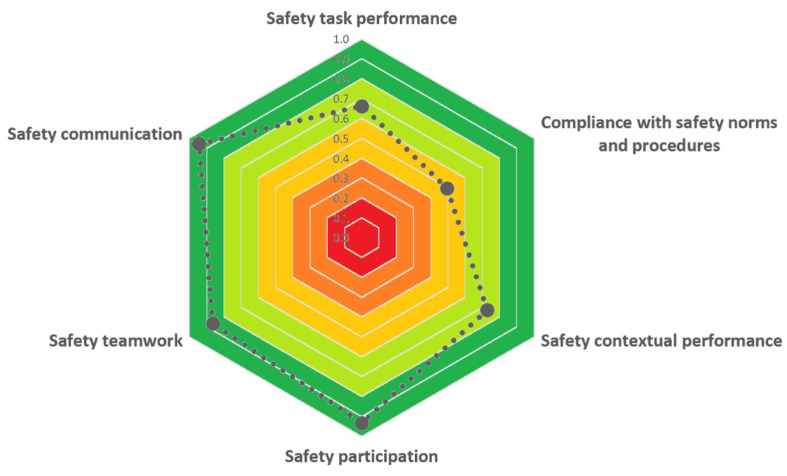
Direct barriers score obtained in the test case.

**Figure 5 ijerph-19-02797-f005:**
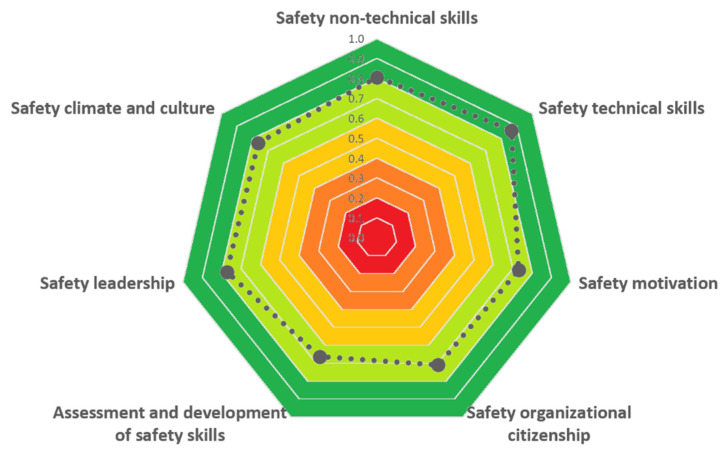
Safeguard barriers score obtained in the test case.

**Table 1 ijerph-19-02797-t001:** Generic task types definition and nominal error probability (NEP).

GTT	Definition	NEP
A	A totally unfamiliar task, performed at speed	0.55
B	Shift or restore the system to a new or original state on a single attempt without supervision	0.26
C	A complex task requiring a high level of comprehension and skill	0.16
D	Routine, highly practiced, a rapid task involving a relatively low level of skill	0.02
E	Restore or shift a system to original or new state following procedures, with some checking	0.003
F	Respond correctly to system command even when there is an augmented or automated supervisory system	2 × 10^−5^

**Table 2 ijerph-19-02797-t002:** Definition of the PSFs included in the model.

PSF	Definition
Available time	The amount of time an operator or a crew has to diagnose and act to perform a task.
Threat Stress	The dangerousness of the task in terms of psychological and physical harm and operators’ adequacy of risk perception.
Complexity	How difficult the task is to perform in the given context.
Experience/Training	The experience and training of the operator(s) involved in the task. Included in this consideration are the years of experience of the individual or crew, and whether or not the operator/crew has been trained on the type of accident, the amount of time passed since training and the systems involved in the task and scenario.
Procedures	The existence, the quality and actual use of formal operating procedures for the tasks under consideration.
Human Machine Interfaces (HMI)	The equipment, displays and controls, layout, quality, and quantity of information available from instrumentation, and the interaction of the operator/crew with the equipment needed to carry out tasks.
Environmental Context	Physical conditions of the working environment (e.g., noise, temperature, brightness) in which the task is performed.
Fatigue	Whether or not the individual performing the task is physically and mentally fit to perform the task at the time. Things that may affect fitness include fatigue or sickness.

**Table 3 ijerph-19-02797-t003:** PSF levels and multipliers.

PSF	Level	Multiplier
Available Time	Inadequate time	100
	Barely adequate time	50
	Limited time	10
	Nominal	1
	Extra Time	0.1
Threat Stress	Very threatening	5
	Moderately threatening	2
	Nominal	1
Complexity	Overly complex	50
	Moderately complex	10
	Nominal	1
	Simplified task	0.1
Experience/Training	The mismatch between knowledge or skills and correct behavior	100
	No experience/training	50
	Low experience/training	15
	Nominal	1
	Extensive experience/training	0.1
Procedures	No procedures available or not used	50
	Very poor procedures	20
	Poor procedures	5
	Adequate and followed procedures	1
	Exceptionally good and followed procedures	0.5
HMI	Completely inadequate	100
	Inadequate	50
	Barely adequate	10
	Adequate	1
	Specifically designed to ease the task	0.5
Environmental Context	Environmental conditions do not allow to perform the task	100
	Adverse conditions	10
	Nominal	1
Fatigue	High	100
	Moderate	10
	Nominal	1

**Table 4 ijerph-19-02797-t004:** Human barriers dimensions and definitions.

Barrier Type	Dimension	Definition
Direct Barriers	Safety task performance	The quality of the operator’s performance in terms of safety in relation to a given task.
	Compliance with safety norms and procedures	The set of activities that an operator needs to carry out to maintain high levels of safety while performing a task. It involves complying with safety norms, procedures and standards, using PPE.
	Safety contextual performance	The set of behaviors that grant the development of a safe environment at work. An example of performance is the operator’s active participation in safety briefings or safety cooperation behaviors among workers.
	Safety participation	Workers’ proactivity and efforts toward the improvement of safety at work and safety performances.
	Safety teamwork	Operators’ ability to work in teams pursuing goals while safely performing tasks.
	Safety communication	Quality and quantity of relevant safety information exchanges about safety in working teams.
Safeguard Barriers	Non-Technical Safety skills	The set of cognitive, social and personal resource skills complementing technical skills and contributing to safe and efficient task performance (e.g., fatigue and stress management, safety awareness).
	Technical Safety Skills	The set of technical skills that operators need to own to work safely. These skills vary as a function of the task to be performed and the operators’ role.
	Safety Motivation	Refers to workers’ willingness to spend energies and efforts to work safely. The worker may be intrinsically motivated to engage in safe behaviors or extrinsically motivated by external pressures from the organization.
	Safety organizational citizenship	Enlargement of workers’ role about safety without a formal acknowledgement of the acquired functions from the organization (e.g., through rewards).
	Assessment and development of safety skills	Evaluates the quality of organizational management of workers’ skills. It mainly focuses on how and how often organizations assess workers’ safety skills and their commitment to improving these.
	Safety Leadership	Evaluates the quality of members-leader interactions about safety. It involves the ability of leaders to promote safety behaviors among workers and improve the overall levels of safety in the working environment.
	Safety Climate and Culture	Refers to the set of workers’ perceptions about safety rules, norms and procedures, and the importance of safety in the working environment.

**Table 5 ijerph-19-02797-t005:** Indicators levels and scores.

Level	Score
Yes	1.00
Rather Yes than No	0.66
Rather No than Yes	0.33
No	0

**Table 6 ijerph-19-02797-t006:** Conversion coefficient for direct barriers score (*CCDB*).

DBS	Conversion Coefficient
DBS ≥ 0.75	0.2
0.55 ≤ DBS 0.75	0.6
0.35 ≤ DBS 0.55	1.0
0.15 ≤ DBS 0.35	1.4
DBS 0.15	1.8

**Table 7 ijerph-19-02797-t007:** Conversion coefficient for safeguard barriers score (*CCSB*).

SBS	Conversion Coefficient
SBS ≥ 0.75	0.2
0.55 ≤ SBS 0.75	0.6
0.35 ≤ SBS 0.55	1.0
0.15 ≤ SBS 0.35	1.4
SBS 0.15	1.8

**Table 8 ijerph-19-02797-t008:** Test case PSF evaluation.

PSF	Selected Level	Multiplier
Available time	Extra Time	0.1
Threat Stress	Nominal	1
Complexity	Nominal	1
Experience/Training	Extensive experience/training	0.1
Procedures	Poor procedures	5
Human Machine Interfaces (HMI)	Specifically designed to ease the task	0.5
Environmental Context	Adverse condition	10
Fatigue	Nominal	1

**Table 9 ijerph-19-02797-t009:** Assessment of the indicators of compliance with safety norms and procedures.

Indicator	Evaluation	Score
Do workers pay attention and comply with safety norms and procedures even if they are not easy to apply?	Rather No than Yes	0.33
Do workers use the correct procedures to work safely?	Rather No than Yes	0.33
Do workers correctly use PPE in line with safety norms and procedures?	Yes	1.00
Do workers ensure the maximum respect of safety norms and procedures while working?	Rather No than Yes	0.33
Barrier total score		0.50

**Table 10 ijerph-19-02797-t010:** Calculation of *HEP* variables.

Variable	Score
Nominal Error Probability (NEP)	0.02
PSF Composite (PSFC)	0.05
Conversion Coefficient Direct Barriers (CCDB)	0.2
Conversion Coefficient Safeguard Barriers (CCSB)	0.2

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
