# Peer review of "Integrating Human Barriers in Human Reliability Analysis: A New Model for the Energy Sector"

_ijerph, 2022, doi:10.3390/ijerph19052797_

Round 1

Reviewer 1 Report

Thank you for the opportunity to review the manuscript titled "Human barriers and human reliability analysis integration: The development of a new model in the energy company". This paper presents an integrated model to assess human reliability. It is evident that a lot of effort has been put into the study to review prior models and to develop a new methodology. That said, I believe that there are several areas where the paper could be improved:

(1) Revise the title: Consider revising the title of the paper- the title is too long and confusing- particularly the phrase "the energy company". When I first read the title, it left me wondering- which energy company? Do you mean just one particular firm? Or firms that share certain characteristics? Or the whole industry? As you can see, just from the title itself, the reader is likely to get confused. 

(2) Introduction: I like the fact that your introduction clearly describes the gap in extant literature, and outlines how your study seeks to address this gap. However, the third paragraph in your introduction needs to be revised. Consider adding a "topic sentence" at the start of the paragraph that streamlines the transition (for e.g. "Existing HRA models do not adequately consider psychosocial barriers" that will help the reader know what the paragraph is all about). In addition, you need to change some of the wording in the last paragraph in the introduction- such as the terms "coherent" and "the energy company" (perhaps you mean "a representative energy company"?)

(3) Background section titled "State of the art": Please replace the term "state of the art" with "background" or "Literature review". The term "state of the art" typically refers to the most recent stage in the development of a certain concept/ product, etc., but in this section, you describe history going 30 years back. With so much historical information in here, this section cannot be categorized under the phrase "state of the art". You might also consider abridging this section by keeping the most relevant information. From my reading of the rest of the paper, there's a lot of redundant information here. 

(4) Methodology: This section is long and confusing. First, you need to start with your selection of this multinational energy company as your empirical context. Why this company? What are the characteristics of its workforce? Are these characteristics easily generalizable to other contexts? In addition, your methodology section should describe what you are doing in the current study. The works of other scholars and the descriptions/ justifications, etc. should be moved to a different subsection. I liked what you have on page 9- this section should be moved further up. By reading the methodology section, the reader should be able to clearly trace your research steps, which is lost right now in the long descriptions in the text. 

Author Response

Responses to Reviewers

Reviewer #1:

Thank you for the opportunity to review the manuscript titled “Human barriers and human reliability analysis integration: The development of a new model in the energy company”. This paper presents an integrated model to assess human reliability. It is evident that a lot of effort has been put into the study to review prior models and to develop a new methodology. That said, I believe that there are several areas where the paper could be improved:

(1) Revise the title: Consider revising the title of the paper- the title is too long and confusing- particularly the phrase “the energy company”. When I first read the title, it left me wondering- which energy company? Do you mean just one particular firm? Or firms that share certain characteristics? Or the whole industry? As you can see, just from the title itself, the reader is likely to get confused. 

Response: Thank you for your comment. We do agree that the title was needlessly long and confusing. We changed it to: “Integrating Human Barriers in Human Reliability Analysis: a New Model for the Energy Sector”. We hope that the new title is sufficiently informative and clear, whilst being shorter.

(2) Introduction: I like the fact that your introduction clearly describes the gap in extant literature, and outlines how your study seeks to address this gap. However, the third paragraph in your introduction needs to be revised. Consider adding a “topic sentence” at the start of the paragraph that streamlines the transition (for e.g. “Existing HRA models do not adequately consider psychosocial barriers” that will help the reader know what the paragraph is all about).

Response: Thank you for your suggestion. We added a “topic sentence” at the start of the 3rd paragraph in the introduction, following your recommendation.

In addition, you need to change some of the wording in the last paragraph in the introduction- such as the terms “coherent” and “the energy company” (perhaps you mean “a representative energy company”?)

Response: We agree that some terms in the last paragraph of the introductions were not appropriate. We apologize for the mistake. We changed the wording according to your suggestion.

(3) Background section titled “State of the art”: Please replace the term “state of the art” with “background” or “Literature review”. The term “state of the art” typically refers to the most recent stage in the development of a certain concept/ product, etc., but in this section, you describe history going 30 years back. With so much historical information in here, this section cannot be categorized under the phrase “state of the art”. You might also consider abridging this section by keeping the most relevant information. From my reading of the rest of the paper, there’s a lot of redundant information here. 

Response: Thank you for your comment, we changed the title of the sections to “Literature Review”, as you suggested. We tried to abridge the section as well. However, we feel that most of the information in this section are relevant for explaining the development of the model for the reader, so the deleted parts are minimal. If the editor and the reviewers feel that further abridging this section is necessary, we will shorten it.

(4) Methodology: This section is long and confusing. First, you need to start with your selection of this multinational energy company as your empirical context. Why this company? What are the characteristics of its workforce? Are these characteristics easily generalizable to other contexts?

Response: We do agree that that information is crucial. We added a paragraph right after the description of the company characteristics, which explains why we chose that specific organization and why it can be considered, to a large extent, representative of the energy sector.

 In addition, your methodology section should describe what you are doing in the current study. The works of other scholars and the descriptions/ justifications, etc. should be moved to a different subsection. I liked what you have on page 9- this section should be moved further up. By reading the methodology section, the reader should be able to clearly trace your research steps, which is lost right now in the long descriptions in the text. 

Response: Thank you for pointing this out. We do agree that the reader would expect a different content in a section named “Methodology”. However, this section describes to the readers all the crucial steps that have been carried out to adapt previous HRA models, integrating human barriers, with the development of a new model as the final output. This means that this section is the core of the present work, and if we removed the descriptions and justification in relation to other scholars’ work, we would omit essential information for the methodology development process. In particular, we would miss the logical steps that led to the improved and integrated methodology. Thus, being the present work about the methodology development, we changed the section’s title accordingly and added an introductory paragraph to specify it. We are glad you liked what we have on page 9, which is a recap of how the assessment process in the model works, but we believe that moving it further up would increase confusion in the reader. However, if the editor and the reviewers are sure that this would increase the paper’s readability, we are available for making the suggested changes.

Reviewer 2 Report

The literature reviewed in this study is to a greater extent sufficient to explain the different aspects of Human Barriers and Human Reliability Analysis integration. However there are missing links in the literature concerning how past models in the energy company have performed in relation to analyzing human barriers and human reliability.

 Therefore a more extensive model is ideal and this has to focus on extending the model to include a variety of psychological elements as safety obstacles. Furthermore, the estimate of HEP must be based on mathematical methodologies that have been verified. When the approach is applied to bigger samples, it will be easier to assess the influence of each barrier on HEP. 

Author Response

Responses to Reviewers

Reviewer #2:

The literature reviewed in this study is to a greater extent sufficient to explain the different aspects of Human Barriers and Human Reliability Analysis integration. However there are missing links in the literature concerning how past models in the energy company have performed in relation to analyzing human barriers and human reliability.

Response: Thank you for your comment. Although some reference about how past models have been used in the energy sector are present in the manuscript, we added a few more at line 77.

 Therefore a more extensive model is ideal and this has to focus on extending the model to include a variety of psychological elements as safety obstacles. Furthermore, the estimate of HEP must be based on mathematical methodologies that have been verified. When the approach is applied to bigger samples, it will be easier to assess the influence of each barrier on HEP. 

Response: We do agree with your comment. We have highlighted this in the “Limitations and Further Reaserch” section at the end of the manuscript.

Round 2

Reviewer 1 Report

N/A